# Substance use patterns among individuals with problematic pornography use: A scoping review

Cerina Dubois[1,2]*, Elizabeth C. Danielson[3], Tim Schwirtlich[4], Molly Beestrum[1,4], Dean T. Eurich[2]

1 Department of Mental Health, Bloomberg School of Public Health, Johns Hopkins University, Baltimore, Maryland, United States of America, 2 School of Public Health, University of Alberta, Edmonton, Alberta, Canada, 3 Department of Surgery, University of Chicago, Chicago, Illinois, United States of America, 4 Feinberg School of Medicine, Northwestern University, Chicago, Illinois, United States of America

* cdubois6@jh.edu

## Abstract

Pornography consumption is a highly prevalent behavior in developed countries, with studies indicating that up to 70% of men and 40% of women have viewed pornography within the past year. Substance use in individuals with problematic pornography use (PPU) is an area that warrants further investigation due to the potential for its compounded negative effects including risk for addiction and potential negative effects on mental and emotional health. This scoping review examines substance use patterns and associations among individuals with PPU; and to summarize the different operationalizations of PPU. This is one of the first reviews to evaluate the empirical research on this relationship. Four primary databases were used to conduct the search: MEDLINE (PubMed), Embase, CINAHL, and PsycINFO, up to December 4, 2024. Included studies investigated the association between substance use and PPU or chronic pornography use. After deduplication, 949 references were retrieved, however, only 8 cross-sectional studies were eligible for this review. Substance use classification varied across studies and included: alcohol, smoking, cocaine, substance use disorder, cannabis use disorder, and general drug use. Alcohol was the most frequently studied substance. Although alcohol, cocaine use, and smoking were positively correlated with PPU, the findings were not significant for other substances or substance use disorders. Studies used a variety of PPU definitions. The Problematic Pornography Use Scale was the most frequently used measure to capture PPU. This review suggests there is a large knowledge gap in understanding the intersection between PPU and substance use in both the volume of literature available and a lack of standardization of measuring PPU. Indeed, PPU is currently not officially recognized as a stand-alone disorder in the DSM-5. Longitudinal studies utilizing a consistent definition and measure for PPU are warranted to fully understand its association with each substance use type. **Open Science Framework Registration:** https://doi.org/10.17605/OSF.IO/86X34

**Data availability statement:** All data is freely available to other researchers in the Open Science Framework link: https://doi.org/10.17605/OSF.IO/86X34.

**Funding:** The authors received no specific funding for this work.

**Competing interests:** The authors have declared that no competing interests exist.

**Abbreviations:** AUD, Alcohol Use Disorder; AUDADIS-IV, Alcohol Use Disorder Associated Disabilities Interview Schedule-IV; AUDIT, Alcohol Use Disorders Identification Test; BASIS-24, Behavioral and Symptom Identification Scale; BPS, Brief Pornography Screen; CBD, cannabidiol; CDC, Centers for Disease Control and Prevention; CSBD-19, Compulsive Sexual Behavior Disorder Scale; CUD, Cannabis Use Disorder; CUDIT-R, Cannabis Use Disorders Identification Test-revised; DSM-5, Diagnostic and Statistical Manual of Mental Illnesses; GHB/GBL, gamma-hydroxybutyrate/gamma-butyrolactone; IP, Internet pornography; IP-CRIT, Internet Pornography Addiction; OR, odds ratio; OSF, Open Science Framework; OUD, Opioid Use Disorder; PIU, Problematic Internet Use; PPCS-6, Problematic Pornography Consumption Scale; PPU, Problematic Pornography Use; PPUS, Problematic Pornography Use Scale; PUD, Pornography Use Disorder; PRISMA-ScR, Preferred Reporting Items for Systematic Reviews and Meta-Analyses for Scoping Reviews; SD, standard deviation; SUD, Substance Use Disorder; US, United States

## Introduction

Over 2.5 million people visit pornography websites every 60 seconds [1]. In a recent United States (US) study, 42% of youth between the ages of 10–17 had viewed pornography - with up to 68% of adolescents reporting lifetime exposure to pornography [2]. In the adult US population, up to 11% of males and 3% of females reported feeling like they were addicted to pornography and 1 in 8 males using pornography expressed needing professional help due to compulsive use [3]. With rising internet access and screen time, the prevalence of problematic pornography use (PPU) has also increased [3,4]. The intersection of substance use and behavioral addictions is an emerging area of interest in public health. Among behavioral addictions, specifically addictive digital behaviors, the overconsumption of pornography has become an increasing public health problem [5].

In the literature, PPU is typically defined as an individual's inability to control the use of pornography, leading to detrimental health consequences [6] such as negative impacts on sexual development in adolescence, difficulty controlling sexual impulses, significant disruptions in one's personal life (e.g., impact on sexual relationships and psychological health) [1,5]. PPU has also been associated with a higher likelihood of developing anxiety, stress, depression, and suicidal ideation [7]. Although PPU is not clinically recognized as a stand-alone behavioral disorder (in the Diagnostic and Statistical Manual fifth Editions (DSM-5-TR)), it was recently included as one of the impulse control manifestations under the Compulsive Sexual Behaviour Disorder (CSBD) under the 11th edition of the International Statistical Classification of Diseases and Related Health Problems [8].

In addition, substance use among adolescents and young adults continues to be a public health concern, particularly after the COVID-19 pandemic [9]. Both PPU and substance use (and subsequent substance use disorders) can lead to significant disruptions in daily functioning and overall quality of life [10]. Indeed, PPU may operate similarly to substance use behaviors by stimulating similar pathways in the brain's reward system, which can lead to dependency and escalating consumption [11]. Thus, substance use in individuals with PPU is an area that warrants further investigation due to the potential for its compounded negative effects. As PPU is frequently associated with impulsivity, dysfunction in mental health, and other psychiatric symptoms [12], these underlying risk factors may exacerbate substance use patterns and potentially, an individual's treatment outcomes [13]. Notably, several studies indicate CSBD shares core elements with substance use disorders (SUD); and that these core elements may lead to a higher likelihood of developing a SUD or have co-occurring PPU [14]. Although associations between CSBD and substance use have been reported, a review on the association between substance use and PPU remains absent.

By summarizing and characterizing substance use patterns in PPU individuals, the findings from this review can inform the development of effective treatment approaches to reduce the prevalence and impact of PPU and SUDs, especially for vulnerable adolescent populations. Hence, the objective of this scoping review is to

examine substance use patterns and associations among individuals with PPU or chronic pornography use, hereafter referred to as PPU. Notably, given the early stage of this research and clinical attention, this scoping review will also capture the different operationalizations of PPU in the existing literature.

## Materials and methods

For this review, the Preferred Reporting Items for Scoping Reviews (PRISMA-ScR) checklist (S1 File) was used. Pilot searches were completed on an iterative basis (to ensure we captured all operationalizations of PPU) starting early 2024 and the final search terms were finalized in December 2024. This included peer-reviewed, empirical studies published on and before December 4, 2024. Included studies focused on substance use for individuals reporting any type of problematic pornography use or chronic pornography use and the Center for Disease and Control (CDC)'s defined substances (see section below).

*Protocol and Registration:* The following scoping review has been registered to the Open Science Framework (OSF) - https://doi.org/10.17605/OSF.IO/86X34

***Problematic Pornography Use (PPU)***: PPU or compulsive pornography use is characterized by an individual's inability to control the use of pornography, leading to detrimental health effects [8,15]. Pornography is typically described as content that showcases sexual material with the main goal of sexually arousing or stimulating the viewer. This can encompass various forms such as visual images, videos, texts, or other media (online or printed). Detrimental health effects of PPU can include both mental and physical effects [16]. Effects include anxiety, depression, and low self-esteem. PPU has been reported to alter the brain's award system and can result in decreased sensitivity to pleasure and dopamine [17]. PPU can negatively impact relationships by reducing intimacy and its overuse has been associated with sexual dysfunction [3]. Although PPU is not officially recognized as a disorder in the DSM-5, it is a manifestation of CSBD, which include compulsive use, loss of control, tolerance, and withdrawal symptoms [18]. Individuals with PPU struggle with co-occurring disorders such as depression, anxiety, and substance misuse [5].

For this scoping review, PPU is differentiated from the DSM-5 Compulsive Sexual Behavior Disorder (CSBD) (ICD-11), which is a persistent pattern of failure to control intense, repetitive sexual impulses or urges, resulting in repetitive sexual behavior. Although CSBD could include PPU, this review is not examining substance use trends in individuals with only noted CBSD - as CSBD includes a wide range of sexual behaviors; whereas PPU could be considered a subtype of CSBD, focusing solely on the use of pornographic material. In addition, this review is not examining substance use trends in individuals with internet addiction (same reasons as CSBD).

*Substance Use:* The CDC's definition of substance use was used, which refers to the consumption of various substances including alcohol, cannabis, hallucinogens, inhalants, opioids (prescription and illegal), sedatives/hypnotics/anxiolytics, stimulants, tobacco (nicotine), and other (unknown substances) [19]. Substance use can also include any method of intake, whether consumed, inhaled, injected, or otherwise absorbed into the body. Types of substance use include alcohol, tobacco (cigarettes, cigars, smokeless tobacco), illicit drugs (heroin, cocaine), cannabis, vaping, and prescription drugs. Substances can be used experimentally, recreationally, for medical purposes, and can also be used problematically (in the case of substance use disorders).

## Inclusion criteria

This included studies that investigated the association of substance use with forms of problematic pornography use, chronic pornography use, or PPU. Included studies examined substance use and its association of PPU as an exposure, outcome, dependent/independent variable, or as a confounder. Since there is no diagnostic definition of PPU in the DSM-5, we included all types of chronic use or problematic use (S1 File). It is acknowledged that Compulsive Sexual Behavior Disorder (CSBD) may overlap with behaviors that are often seen in PPU including a pattern of intense and repetitive

sexual thoughts, urges, or behaviors that are difficult to control. For this scoping review, we only included evidence that specifically examined "pornography use" rather than "sexual behavior" (which can encompass pornography use). Included research studied adolescents [12–17] or adults (18+).

For study types, all evidence available including longitudinal studies, clinical interventions, case studies, self-reported outcomes, screening systems, assessment strategies, and follow-up intervention programs, were included. Systematic reviews, literature reviews, clinical reviews, and scoping reviews were also included. This review only included English language studies or those that may have been translated from their original language. The search did not include a limited timeline or geographical region. Studies also included those that examined trends on substance use for individuals who reported chronic, problematic, or uncontrolled pornography use.

### Exclusion criteria

Studies that did not explicitly state both substance use (which includes any of the above mentioned types of use) and pornography use, were excluded. Excluded studies were animal studies, expert opinion pieces, blogs, editorials, and grey literature from conference proceedings, meeting abstracts, and dissertation theses. Articles that examined addiction in the form of social media, internet use, mobile phone use, online gaming, online gambling (not specific to pornography) were excluded. In addition, excluded articles also were those that solely focused on sex addiction, hypersexuality, and CSBD.

### Information sources

A subject expert librarian, MB, independently chose the search terms for the scoping review. Subsequently, CD and ED were secondary reviewers of these search terms. The most recent search was executed on December 4, 2024 without date limitations.

### Search strategy

Four primary databases were used to conduct the search: MEDLINE (PubMed), Embase, CINAHL, and PsycINFO. The Embase Drug library was selected to capture all generic and standard drug names of medical cannabis that are currently available. The full search terms used can be found in B.

A pilot search was conducted by MB on September 14, 2024, to identify key MeSH terms/words that ensured inclusion of all types of "substance use" and "pornography use." The pilot results were then shared with CD who then independently screened the pilot 100 articles via title and abstract review to determine consistency in screening inclusion and exclusion criteria. MB then released the rest of the articles for CD to conduct the title and abstract screening. CD then shared with ED a list of articles that filled the inclusion criteria.

### Screening process

CD independently screened the titles for each study. The title/abstract screening was conducted via Rayyan to ensure consistency in inclusion/exclusion of articles. The screening protocol included reading the title and abstract followed by answering a list of eligibility questions (S2 File). The results were shared with ED via Rayyan. Both ED and CD conducted an independent screen of abstract and full-text screening. If the title/abstract did not answer the screening questions, the citation was included for the full-text screening process. For included article, CD and ED reviewed the references to capture any missed relevant article from the search. Any discrepancies were discussed between ED and CD to come to consensus on the final number of included articles.

### Data charting and synthesis

Data extraction from the final list of included studies was carried out independently by TS using Microsoft Excel. TS extracted the data, and then CD and ED reviewed the extracted data tables. Once consensus was reached for the final list

of articles, TS extracted the data into tables. The synthesis of the final list of articles in the review was completed by CD, which was then reviewed by ED.

### Data items

Data items included the first author, year, type of study, location/country, sample size, sociodemographic information about the population, intervention or exposure (if applicable), comparison/control, study outcomes, substance use captured, substance use measures, statistical methods, main findings of associations between substance use and PPU, as well as definition and operationalization of PPU.

## Results

### Study selection

After de-duplication, 949 articles were included for initial title and abstract screening. Next, 27 articles were included for full-text screening. After full-text screening, we had a remaining 8 studies included in qualitative synthesis (Fig 1).

### Study and participant characteristics

Sample size ranged from 172 to 1272 participants, Table 1. Studied populations were individuals with substance use disorders (n = 3) [12,14,20], adolescent and young adult students (n = 2) [16,21], veterans (n = 2) [22,23] and sexual minorities (n = 1) [18]. The mean age of participant groups ranged from approximately 15–40 years - with standard deviations of less than 10 years. The groups predominantly consisted of male participants (60–86%), with one male-only study of US veterans [23]; and one study with a majority of female undergraduate students [16]. Two studies examined specifically US veterans [22,23]. The origins of studies were diverse, including the United States [12,22,23], Canada [16], Poland [18], Italy [20], India [21], and Germany [14].

### Study design

All 8 included studies followed a cross-sectional observational design, characterizing a single group of individuals; or comparing groups at a single point in time stratified by various groupings based on exposure, outcome, or comparators (Table 1). For most studies (6 of 8), pornography use was an outcome or variable of interest [12,14,16,18,22,23]; and in the other 2 studies, pornography use was a covariate [20,21]. Half included control groups based on an outcome such as substance or alcohol use disorder [14,22], smoking [21], cannabis [16], other drugs [14,20], and PPU [12]. All studies employed multiple forms of statistical comparative analysis, including chi-square tests, t-tests, correlation analyses, and regression analyses.

### Type of substance use measured

Substance type varied across studies and included: alcohol [12,14,16,22,23], smoking [21], cocaine [20], substance use disorder [14,23], cannabis use disorder [16], and drug use [14,18] (Table 2). Alcohol was the most frequently studied substance type (5 of 8) [12,14,16,20,22,23]. Alcohol use disorder or dependence (AUD) was measured in two studies. [22,23]. Golder et al. (2024) [14] was the only study that examined polysubstance use and assessed whether the individual used at least one of the following substances: alcohol, cannabis, cocaine, amphetamine, opioids, hallucinogens, ecstasy, benzodiazepines, and ketamines. Lewczuk et al. (2014) [18] was the only study that measured 'chemsex' drugs, which included psychoactive substances to prolong, enhance, or sustain sexual activity. This included methamphetamine, mephedrone, GHB/GBL (gamma-hydroxybutyrate/gamma-butyrolactone), or ketamine.

### Pornography operationalization and measures

Across the included studies, no consistent or standard definition of PPU was used (Table 3). Although pornography is not considered a stand-alone DSM-5 recognized disorder, the term, 'pornography addiction' was used in several studies. All

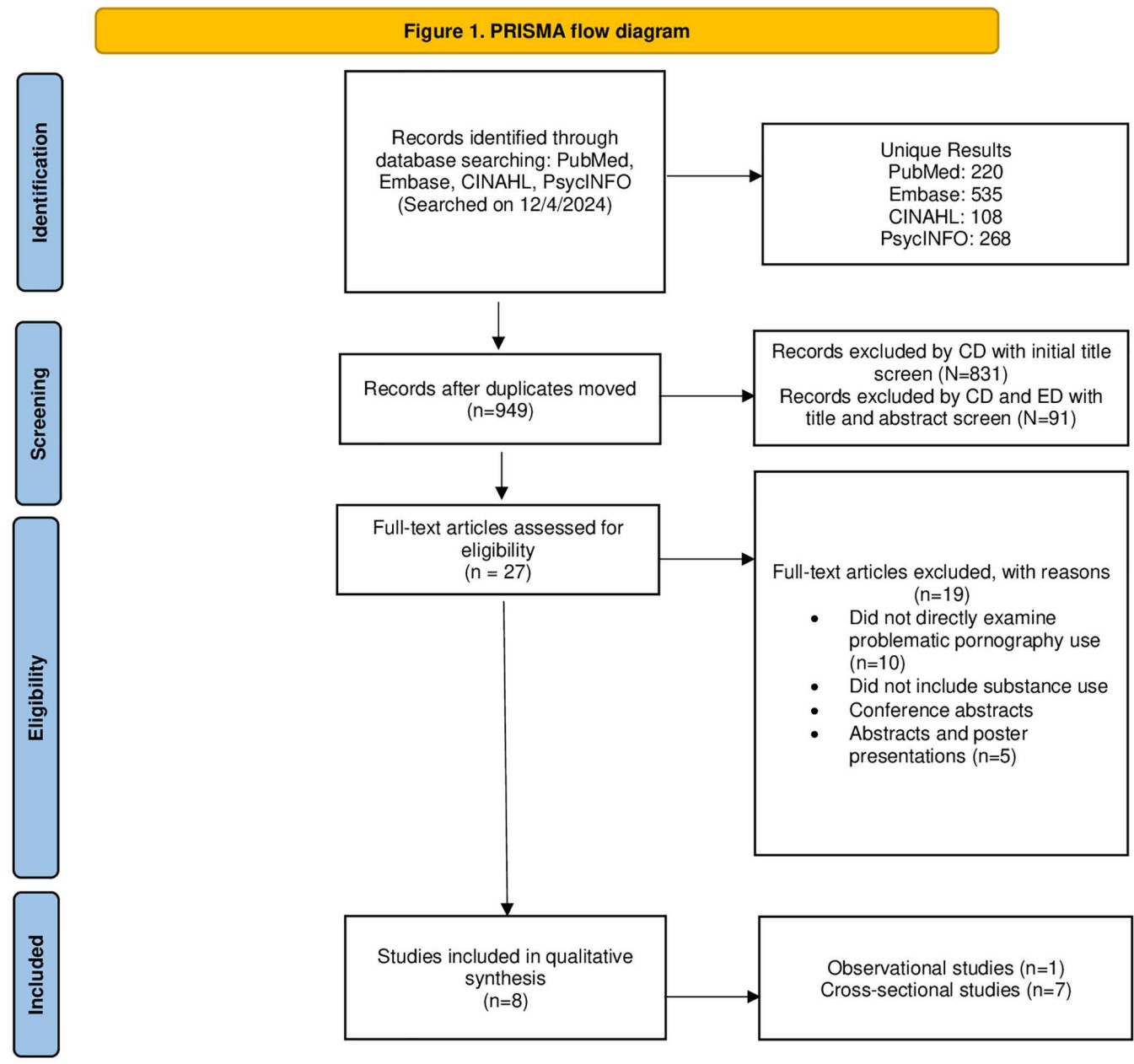

**Figure 1. PRISMA flow diagram**

*From:* Page MJ, McKenzie JE, Bossuyt PM, Boutron I, Hoffmann TC, Mulrow CD, et al. The PRISMA 2020 statement: an updated guideline for reporting systematic reviews. BMJ 2021;372:n71. doi: 10.1136/bmj.n71

**Fig 1. PRISMA flow diagram.**

studies captured PPU through some form of a self-reported questionnaire. The most common definition of PPU was as a behavioral habit leading to negative consequences and distress, often accompanied by control or impulse difficulties. Of the four studies following this definition, two studies [22,23] utilized the Problematic Pornography Use Scale (PPUS),

**Table 1. Study characteristics.**

| | Author* Alphabetical order | Year | Study Setting (location, city, country) | Study Design | Sample Size | Population | Participant Demographics (age, sex) | Intervention or Exposure | Comparator/ Control (Y/N) | Outcome(s) | Was pornography use a primary outcome or independent variable of interest in the study? (Y/N) | Statistical Method |
|---|---|---|---|---|---|---|---|---|---|---|---|---|
| 1 | Bagchi | 2014 | Six co-educational higher secondary schools of Kolkata (India) | Cross-sectional, observational study | 526 | Students of 15–19 years | Age range: 15-19 years Smokers (n = 156): 134 (86%) male, 22 (14%) female Non-smokers (n = 370): 229 (62%) male, 141 (38%) female | Family, peer group, and personal factors, including | Y Compared smokers versus non-smokers | Smoking | N | Chi-square test |
| 2 | Baroni | 2019 | Territorial outpatient services for drug-addicted individuals (SERTs) in the Calabria region (Italy) | Cross-sectional observational study | 183 | Drug-addicted individuals | Adults (age not reported), 148 (81%) male, 35 (19%) female | Characteristics of internet use, including addiction to pornography | N | Problematic internet use (PIU) | N | Chi-square test, t-test |
| 3 | Golder | 2024 | Inpatient addiction and psychosomatic clinics at Salus Kliniken Friedberg and Friedrichsdorf (Germany) + commercial online survey for control group | Cross-sectional observational study | 184 | SUD inpatient + general population (control) | SUD (n = 92): mean age 39.9 years (SD = 10.3), 59 (64%) male, 33 (36%) female No SUD (n = 92): mean age 39.3 years (SD = 10.5), 60 (65%) male, 32 (35% female) | Compulsive sexual behavior disorder (CSBD) and subtype pornography use disorder (PUD) | Y Compared Individuals with SUD versus no SUD | Substance use disorder (SUD) | Y | Chi-square test, Welch t-test |
| 4 | Harper | 2016 | University of Calgary Research Participation System (Canada) | Cross-sectional observational study | 191 | Undergraduate students | Mean age 21.05 years (SD = 2.96, range: 17–38), 86 (45%) male, 105 (55%) female | Psychosocial functioning and addictive propensities (including alcohol and cannabis) | N | Problematic Internet pornography (IP) use and IP addiction | Y | Bivariate Pearson correlations, independent t-test, regression analysis |
| 5 | Lewczuk | 2024 | Online research platform (Poland) | Cross-sectional observational study | 198 | Cisgender sexual minorities | Mean age 27.13 years (SD = 7.78), 144 (73%) male, 54 (27%) female | Sexual minority stress, social support, and sexualized drug use | N | Compulsive sexual behavior disorder (CSBD) and problematic pornography use (PPU) | Y | Pearson correlation, regression analysis |

*(Continued)*

**Table 1.** (Continued)

| | Author* Alphabetical order | Year | Study Setting (location, city, country) | Study Design | Sample Size | Population | Participant Demographics (age, sex) | Intervention or Exposure | Comparator/ Control (Y/N) | Outcome(s) | Was pornography use a primary outcome or independent variable of interest in the study? (Y/N) | Statistical Method |
|---|---|---|---|---|---|---|---|---|---|---|---|---|
| 6 | Moisson | 2019 | Secondary analysis of data from Survey of the Experiences of Returning Veterans, telephone interviews + web-survey (US) | Cross-sectional analysis of prospective longitudinal observational study | 283 | US veterans | Mean age 35.1 years (SD = 9.2) Lifetime history of AUD (n = 109): 84 (77%) male, 25 (23%) female No lifetime history of AUD (n = 170): 113 (66%) male, 57 (34%) female | Psychopathology and hypersexuality factors, including problematic use of pornography | Y Compared lifetime history of AUD versus no lifetime AUD | Lifetime alcohol-use disorder (AUD) | Y | Chi-square test, Welch t-test, regression analysis |
| 7 | Shirk | 2021 | Secondary analysis of data from Survey of the Experiences of Returning Veterans (US) | Cross-sectional analysis of prospective longitudinal observational study | 172 | Male US veterans (ever watching pornography) | Mean age 33.9 years (SD = 8.53), male only | Demographic, psychiatric, and substance use comorbidities | N | Problematic pornography use (PPU) | Y | Bivariate analysis, regression analysis |
| 8 | Stefanovics | 2024 | APT Foundation for treatment of substance use disorder, headquartered in New Haven, Connecticut (US) | Cross-sectional observational study | 1272 | Individuals seeking opioid use disorder (OUD) treatment | PPU (n = 57): mean age 39.31 years (SD = 8.47), 40 (77%) male, 17 (23%) female No PPU (n = 1215): mean age 37.71 years (SD = 10.5), 667 (60%) male, 548 (40%) | Impulsive tendencies and psychiatric symptoms, including substance abuse | Y Compared individuals with PPU versus no PPU | Problematic pornography use (PPU) | Y | Chi-square test, t-test, regression analysis |

**Legend: AUD:** Alcohol Use Disorder, **BPS**: Brief Pornography Screen, **CSBD-19**: Compulsive Sexual Behavior Disorder Scale, **CUDIT-R**: Cannabis Use Disorders Identification Test-revised, **IP:** Internet pornography, **OR**: odds ratio, **OUD**: Opioid use disorder, **PIU**: Problematic Internet Use, **PPU**: Problematic Pornography Use **PUD**: Pornography Use Disorder, **SD:** standard deviation, **SUD:** Substance Use Disorder, **US:** United States.

while the other two [12,18] employed the Brief Pornography Screen (BPS). The PPUS is a 12-item instrument that uses a 6-point Likert scale, referencing a previous 12-month time frame. As indicated in the name, the BPS is a shorter version for screening pornography use within the last six months on a 5-item, 3-point Likert scale.

Golder et al. (2024) [14] measured pornography use disorder (PUD), characterized as the combination of CSBD and PPU within the last six months, with a short version of the Problematic Pornography Consumption Scale (PPCS-6). It is a 6-item questionnaire with a 7-point Likert scale, where PPU is defined as a cumulative score of 20 or greater, in combination with a CSBD-19 score of 50 or greater. The remaining three studies did not further specify their measured concept of "PPU". Moreover, the three studies utilized self-designed items [20,21] or adapted items from an internet gaming disorder measurement instrument [14] as part of their questionnaire, attributed to retrieving quantitative PPU scores.

**Table 2. Synthesis of results and measures of substance use.**

| | Author *Alpha-betical order | Year | Types of Substance Use Captured in relation to PPU | Measures to Capture Substance Use | Polysub-stance use (Y/N) | Main Findings (with confidence intervals) |
|---|---|---|---|---|---|---|
| 1 | Bagchi | 2014 | Smoking behavior | Self-reported questionnaire | N | Significant correlation between smoking and pornography addiction<br>• OR: 3.1 [2.1, 4.56]<br>• p-value: < 0.0001 |
| 2 | Baroni | 2019 | Cocaine | Self-reported questionnaire (Likert five-point scale) | N | Significant correlation between cocaine use and pornography addiction<br>• Scores: 3.59 ± 1.44 vs 2.54 ± 0.41<br>• p-value: < 0.001 |
| 3 | Golder | 2024 | Use of at least one of the following substances: alcohol, cannabis, cocaine, amphetamine, opioids, methamphetamine, halluci-nogens, ecstasy, benzodi-azepines, and ketamines | Clinical diagnosis based on the Alcohol Use Disorders Identification Test (AUDIT), a questionnaire for prescription drug misuse, Fagerström Test for Nicotine Dependence, and the Inventory of Drug-Taking Situa-tions + clinical interview and diagnosis | Y | Non-significant group differences for PUD prevalence between SUD and control:<br>• Chi-square: 2.080, p-value: 0.149<br>Adversely significant problematic pornography use scores between SUD and control:<br>• Mean (control): 14.13 (SD = 8.36)<br>• Mean (SUD): 10.76 (SD = 6.15)<br>• t-test: 3.03<br>• p-value: 0.003 |
| 4 | Harper | 2016 | Alcohol, cannabis | Alcohol Use Disorders Identifica-tion Test (AUDIT), Cannabis Use Disorders Identification Test – revised (CUDIT-R) | N | Non-significant correlations between:<br>• AUDIT and IP addiction: 0.049 (p-value > 0.05)<br>• CUDIT-R and IP addiction: 0.125 (p-value > 0.05)<br>(Significant correlations of both AUDIT and CUDIT-R with frequency of pornography use) |
| 5 | Lewczuk | 2024 | Sexualized drug use or "chemsex" (psychoactive substances to prolong, enhance or sustain sexual activity, i.e., methamphet-amine, mephedrone, GHB/GBL, or ketamine) | Question about how often in the past 12 months the person had engaged in sexual activities under the influ-ence of psychoactive substances to make sexual activity easier or last longer, or to enhance the sexual experience | N | Non-significant correlation and regression coefficient for frequency of sexualized drug use on PPU symptoms:<br>• Correlation: 0.00, p-value > 0.05<br>• Beta-coefficient: 0.02 [-0.12, 0.17], p-value: 0.744 |
| 6 | Moisson | 2019 | Alcohol dependence/ Alco-hol Use Disorder (AUD) | Alcohol Use Disorder Associated Disabilities Interview Schedule-IV (AUDADIS-IV) | N | Significant association between PPU and AUD:<br>• t-test: -2.20<br>• p-value: 0.029<br>PPU non-significant predictor in regression model (no metrics provided) |
| 7 | Shirk | 2021 | Alcohol Use Disorder (AUD) [+Substance Use Disorder (SUD)] | Alcohol Use Disorder Associated Disabilities Interview Schedule-IV (AUDADIS-IV) | N | Positive but no significant association between AUD and PPU:<br>• beta-coefficient: 0.14,<br>• t-test: 1.81<br>• p-value: 0.072<br>No statistical analysis because too little sample size for SUD (4.70%) |
| 8 | Ste-fanovics | 2024 | Alcohol/ Drug Use | Self-reported Behavioral and Symp-tom Identification Scale (BASIS-24), with 4 items focused on substance abuse on 5-point Likert scale | N | Significant association between alcohol/drug use and PPU<br>• t-test: 4.02<br>• p-value: < 0.0001 |

**Legend: AUD:** Alcohol Use Disorder, **AUDADIS-IV:** Alcohol Use Disorder Associated Disabilities Interview Schedule-IV, **BASIS-24:** Behavioral and Symptom Identification Scale, **AUDIT**: Alcohol Use Disorders Identification Test, **CSBD-19**: Compulsive Sexual Behavior Disorder Scale, **CUDIT-R**: Can-nabis Use Disorders Identification Test-revised, **GHB/GBL:** gamma-hydroxybutyrate/gamma-butyrolactone, **OR**: odds ratio, **PIU**: Problematic Internet Use, **PPU**: Problematic Pornography Use, **PPUS**: Problematic Pornography Use Scale, **PUD**: Pornography Use Disorder, **SD**: standard deviation, **SUD**: Substance Use Disorder.

PLOS Global Public Health

**Table 3. Operationalization of problematic pornography use.**

| | Author | Year | Operationalization | Definition |
|---|---|---|---|---|
| 1 | Bagchi | 2014 | N/A (self-reported, no disclosure of questionnaire survey items) | Term 'pornography addiction' was used. No definition was provided. |
| 2 | Baroni | 2019 | Two dedicated self-reported questions: "26. Do you connect to the Internet to look for erotic stuff?", "27. Do you prefer the excitement of what you can find online than intimacy with your partner?" (self-reported, 5-point Likert scale) | Term 'pornography addiction' was used. No definition was provided. |
| 3 | Golder | 2024 | Short version of the Problematic Pornography Consumption Scale (PPCS-6) referencing last 6 months (self-reported, 6 items on 7-point Likert scale, PPU defined as PPCS-6 scores >= 20 and Compulsive Sexual Behavior Disorder Scale (CSBD-19 scores >= 50) | Pornography use disorder (PUD) is defined as the combination of CSBD and problematic pornography use (PPU: problematic pornography consumption over the past 6 months) |
| 4 | Harper | 2016 | Adapted DSM-5 (criteria for diagnosing internet gaming disorders) for internet pornography addiction (IP-CRIT) questionnaire referencing last 12 months (self-reported, 14 items on 4-point Likert scale) | Term 'Internet pornography addiction' was used. No definition was provided. |
| 5 | Lewczuk | 2024 | Brief Pornography Screen (BPS) referencing last 6 months (self-reported, 5 items, 3-point Likert scale, PPU defined as BPS scores >= 4) | Problematic Pornography Use (characterized by a preoccupation with pornography to the point that it causes distress and leads to negative consequences, with pornography use also being employed as a possible coping strategy to deal with difficult emotions) |
| 6 | Moisson | 2019 | Problematic Pornography Use Scale (PPUS) referencing last 12 months (self-reported, 12 items on 6-point Likert scale) | Problematic Pornography Use (as per PPUS: problems related to four domains: distress and functional impairment, excessive use, control difficulties, use of pornography to escape/avoid negative emotions) |
| 7 | Shirk | 2021 | Problematic Pornography Use Scale (PPUS) referencing last 12 months (self-reported, 12 items on 6-point Likert scale) | Problematic Pornography Use (as per PPUS: problems related to four domains: distress and functional impairment, excessive use, control difficulties, use of pornography to escape/avoid negative emotions) |
| 8 | Stefanovics | 2024 | Brief Pornography Screen (BPS) referencing last 6 months (self-reported, 5 items on 3-point Likert scale, PPU defined as BPS scores >= 4) | Problematic Pornography Use (referenced different classifications: subtype of sexual disorders vs. independent behavioral addiction) |

**Legend: BPS**: Brief Pornography Screen, **CSBD-19**: Compulsive Sexual Behavior Disorder Scale, **OR**: odds ratio, **PIU:** Problematic Internet Use, **PPU**: Problematic Pornography Use, **PPUS:** Problematic Pornography Use Scale, **PUD:** Pornography Use Disorder.

## Alcohol use

Alcohol use was measured using: the Alcohol Use Disorder Identification Test (AUDIT), the Alcohol Use Disorder Associated Disabilities Interview Schedule-IV (AUDADIS-IV), and the Behavioral and Symptom Identification Scale (BASIS-24). One study used BASIS-24, however, alcohol was listed within subcategories of drug use [12]. In the 2 studies that utilized AUDIT [14,16], both reported non-significant correlations between AUDIT and internet pornography addiction (p>0.05); however, Harper et al.(2016) [16] found statistically significant correlations between AUDIT and frequency of pornography use. In the two studies that utilized AUDADIS-IV, Moisson et al. (2019) [22] reported a significant association between AUD and PPU (t-test: -2.20, p=0.03). Conversely, Shirk et al. (2021) [23] reported a positive (but not significant) association between AUD and PPU (t-test: 1.81, p=0.07). Lastly, Stefanovics et al. (2024) [12] reported a significant association between alcohol/drug use and PPU (t-test: 4.02, p<0.0001).

## All other substances & PPU

*Smoking*: Bagchi et al. (2014) [21] specifically examined smoking and PPU. Using a self-reported questionnaire, the study reported a statistically significant association between smoking and pornography addiction (OR: 3.1; p<0.0001).

***Cocaine:*** Baroni et al. (2019) [20] examined the correlation between cocaine use and pornography addiction. The use of various substances was measured through a Likert 5-point scale in a self-reported questionnaire. In this case, cocaine was the only substance that was reported to have a statistically significant positive correlation with pornography addiction (p<0.001).

***Substance Use Disorder (SUD):*** Golder et al. (2024) [14] examined SUD in relation to PPU, in which SUD was clinically diagnosed based on the AUDIT scores, nicotine dependence scores, and the Inventory of Drug-Taking Situations – in concurrence with a clinical interview. The study found non-significant group differences in PPU between those with SUD versus controls (p=0.14). However, individuals in the control group had unexpectedly significantly higher scores on the PPCS-6 (frequency of problematic pornography consumption) than those with SUD (p=0.003). On the other hand, Shirk et al. (2021) [23] could not report any results for the association between SUD and PPU due to too small of a sample size.

***Cannabis Use Disorder (CUD):*** Harper et al. (2016) [16] examined CUD in relation to PPU, in which CUD was measured through the Cannabis Use Disorders Identification Test-revised (CUDIT-R). The study found a non-significant correlation between CUDIT-R and internet pornography addiction (p>0.05); however, a significant correlation was found between CUDIT-R and the frequency of pornography use.

***Sexualized Drug Use:*** Lewczuk et al. (2014) [18] was the only study to examine sexualized drug use or "chemsex" substances with PPU. The measure of PPU was captured through one survey question that asked about how often in the past 12 months the individual had engaged in sexual activities under the influence of psychoactive substances to make sexual activity easier or last longer. The study did not find a significant correlation for the frequency of sexualized drug use on PPU symptoms (p>0.05).

## Discussion

This scoping review examined the association between PPU and substance use. Despite an increasing trend of PPU in individuals across the globe, this review identified only 8 studies that directly examined the relationship between substance use and PPU, identifying a notable gap in the literature. This limited number of studies available suggests that the intersection between PPU and substance use remains under studied. A large proportion of the studies (5 studies) examined alcohol and PPU, from which 2 reported a statistically significant correlation [12,22]; and 3 not significant [14,16,23]. Smoking and cocaine use were also reported to be significantly associated with PPU [20,21]. However, in the Golder et al. (2024) study [14], individuals in the control group had unexpectedly statistically significantly higher scores on the PPCS-6 (higher problematic pornography consumption) than individuals with SUD (p=0.0003). The researchers attributed these results to underreporting due to potential stigma, differences in relationship status, differing treatment response, differing living environments, and underestimation of specific behaviors related to pornography use. Likewise, another study found insignificant correlations with PPU in relation to CUD and sexualized drug use. From these differing results, mechanisms between PPU and substance use may be more complex and differ pending the type of substance use.

Although both substance use and pornography use are both prevalent public health issues with potential negative health impacts, empirical research surrounding PPU is still in its infancy, especially when compared to more established areas of research such as substance use disorders. Due to preliminary findings and limited research studies, the classification of PPU continues to be a challenge. Furthermore, there is currently no standardized measure of what constitutes 'problematic' pornography use and a lack of validated instruments to assess PPU across diverse populations. As observed in this review, existing studies rely on self-report surveys, which can be influenced by shame, guilt, and stigma, which may lead to underreporting of PPU and acknowledgement of having symptoms of PPU.

This scoping review observed that PPUS was the most frequently used scale in the literature [12,22,23]. However, the review found significant variation in the definitions of PPU, including the use of the terms: 'PUD', 'chronic pornography use', and 'pornography addiction'. In terms of operationalization, at least 3 different scales were used (i.e., PPCS-6, PPUS, and BPS) to measure PPU - with different diagnostic criteria. Furthermore, the overlap with the operationalizations

of IP (IP-CRIT) and CSBD resulted in challenges in interpreting and synthesizing the results across the 8 cross-sectional studies. Indeed, the absence of standardized diagnostic criteria or a best-practice tool in this field hinders the development of a cohesive body of research, which may complicate comparisons across studies.

This scoping review has several implications for research and practice. With the advent of the internet and accessibility to explicit content, the concept of PPU has garnered recent attention. Particularly, in the adolescent population, some studies have indicated that over 90% of young men and a significant proportion of young women regularly view internet pornography and are exposed to pornography at a young age [5]. The 8 cross-sectional studies in this scoping review showed consensus that PPU and substance use often do co-occur, which may suggest that there may be shared underlying mechanisms of addiction, coping strategies, and impulsivity with both behaviors. Both PPU and substance use are associated with negative mental health outcomes, including anxiety, depression, and impaired cognitive function [24]. Given this overlap between PPU and substance use, integrated intervention programs that address both conditions simultaneously could be more effective. Addressing the above gaps warrants ongoing efforts to standardize terminology, classification of PPU, and measurement tools so that researchers can build a robust body of evidence that supports individuals who struggle with both substance use and PPU.

This is the first scoping review to examine the literature on substance use in individuals with PPU. However, the scoping review has several limitations. The review only included 8 cross-sectional studies, which restricted the inferences and conclusions made from the findings. Notably, small sample sizes highly limited the generalizability of the results, and the cross-sectional design did not allow us to examine any causal relationships between PPU and substance use. As mentioned, the variability of how PPU and substance use were captured across studies also contributed to the challenges in cross-comparison and synthesis of the results. Hence, future longitudinal studies should employ are warranted to understand the temporal dynamics and causative factors that link these behaviors.

## Conclusions

This scoping review explored the relationship between PPU and substance use. The review included 8 cross-sectional studies, in which both the operationalizations of PPU and results in its associations with substance use, were mixed in assessing the correlation between PPU and substance use. Future research should focus on standardizing the definition of PPU and conduct longitudinal studies to better understand the causal mechanisms between substance use and PPU.

## Supporting information

**S1 File. List of Eligibility Questions.**
(DOCX)

**S2 File. Search Terms.**
(DOCX)

**S3 File. PRISMA P Checklist.**
(DOCX)

## Author contributions

**Conceptualization:** Cerina Dubois, Elizabeth C. Danielson.

**Data curation:** Molly Beestrum.

**Formal analysis:** Cerina Dubois, Elizabeth C. Danielson, Tim Schwirtlich.

**Funding acquisition:** Cerina Dubois.

**Investigation:** Cerina Dubois, Elizabeth C. Danielson, Tim Schwirtlich, Molly Beestrum.

**Methodology:** Cerina Dubois, Elizabeth C. Danielson, Tim Schwirtlich, Molly Beestrum.

**Project administration:** Cerina Dubois, Elizabeth C. Danielson.

**Resources:** Cerina Dubois, Tim Schwirtlich.

**Software:** Cerina Dubois, Tim Schwirtlich.

**Supervision:** Cerina Dubois, Elizabeth C. Danielson.

**Validation:** Cerina Dubois, Elizabeth C. Danielson, Tim Schwirtlich, Dean T. Eurich.

**Visualization:** Cerina Dubois, Elizabeth C. Danielson, Tim Schwirtlich.

**Writing – original draft:** Cerina Dubois, Elizabeth C. Danielson, Tim Schwirtlich.

**Writing – review & editing:** Cerina Dubois, Elizabeth C. Danielson, Tim Schwirtlich, Dean T. Eurich.

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
