## [Decision Letter · Decision Letter 0]

14 Aug 2025

PGPH-D-25-01648

Substance Use Patterns Among Individuals with Problematic Pornography Use: A Scoping Review

Dear Dr. Dubois,

Thank you for submitting your manuscript to PLOS Global Public Health. After careful consideration, we feel that it has merit but does not fully meet PLOS Global Public Health’s publication criteria as it currently stands. Therefore, we invite you to submit a revised version of the manuscript that addresses the points raised during the review process.

This timely scoping review explores an interesting area which is of general interest to the readers. Please find the excellent suggestions by the reviewer below. 

We look forward to receiving your revised manuscript.

Kind regards,

Susmita Chandramouleeshwaran

Academic Editor

Journal Requirements:

1. Please amend your online Financial Disclosure statement. If you did not receive any funding for this study, please simply state: “The authors received no specific funding for this work.”

2. Please update your online Competing Interests statement. If you have no competing interests to declare, please state: “The authors have declared that no competing interests exist.”

3. In the online submission form, you indicated that “Data extraction supplementary tables can be accessed by request to corresponding author.”. 

a) In a public repository, 

b) Within the manuscript itself, or 

c) Uploaded as supplementary information.

4. Please provide separate figure files in .tif or .eps format only and ensure that all files are under our size limit of 10MB. You may leave the embedded figures in the manuscript.

Additional Editor Comments (if provided):

Reviewers' comments:

Reviewer's Responses to Questions

**Comments to the Author**

1. Does this manuscript meet PLOS Global Public Health’s publication criteria?

Reviewer #1: Yes

2. Has the statistical analysis been performed appropriately and rigorously?

Reviewer #1: N/A

3. Have the authors made all data underlying the findings in their manuscript fully available (please refer to the Data Availability Statement at the start of the manuscript PDF file)?

Reviewer #1: No

4. Is the manuscript presented in an intelligible fashion and written in standard English?

Reviewer #1: Yes

Reviewer #1: I recommend acceptance pending minor revisions to improve consistency in terminology, strengthen interpretive clarity, and expand briefly on practical implications for intervention and policy development.

Full review included as an attachment

**Do you want your identity to be public for this peer review?** For information about this choice, including consent withdrawal, please see our Privacy Policy

Reviewer #1: **Yes: ** Monica Malta

---

## [Editor Report · Decision Letter 1]

30 Sep 2025

Substance Use Patterns Among Individuals with Problematic Pornography Use: A Scoping Review

PGPH-D-25-01648R1

Dear Dr. Dubois,

We are pleased to inform you that your manuscript 'Substance Use Patterns Among Individuals with Problematic Pornography Use: A Scoping Review' has been provisionally accepted for publication in PLOS Global Public Health.

Best regards,

Susmita Chandramouleeshwaran

Academic Editor